# A Novel Preference Elicitation Technique Based on a Graph Model and Its Application to a Brownfield Redevelopment Conflict in China

**DOI:** 10.3390/ijerph16214088

**Published:** 2019-10-24

**Authors:** Shinan Zhao, Haiyan Xu

**Affiliations:** College of Economics and Management, Nanjing University of Aeronautics and Astronautics, Nanjing 211106, China; shinanzhao@nuaa.edu.cn

**Keywords:** graph model for conflict resolution, conflict analysis, preference elicitation, brownfield redevelopment conflicts

## Abstract

Disputes are very common and pervasive in brownfield redevelopment projects, in which multiple stakeholders or decision-makers (DMs) strategically interact with each other with a conflict of interest. The preference information of DMs involved plays a vital role in identifying possible outcomes or resolutions for resolving a tough brownfield conflict. In this research, a novel preference ranking technique is purposefully proposed within the graph model for conflict resolution (GMCR) paradigm to effectively and accurately garner DMs’ actual preferences, in which states are ranked according to their similarities and closeness to the most and least preferred states instead of subjective option statements or weights in traditional preference ranking methods. Finally, a real-world brownfield conflict which occurred in China is utilized to show how the proposed preference ranking method can be applied for conveniently obtaining the true preference information of DMs and strategically determining the equilibria of a given dispute. The case study indicates that the novel preference elicitation approach is more objective and reasonable than the traditional option prioritization method. Moreover, there exists an equilibrium which can provide strategic advice and meaningful insights for addressing the brownfield conflict.

## 1. Introduction

Brownfields refer to idled, vacant, abandoned, or underused industrial and commercial properties in urban areas which may be previously contaminated by chemical substances or toxic pollutants [1,2,3]. The redevelopment and restoration of brownfields have positive impacts on the sustainable development of the economy, society, and environment in a community such as cleaning up hazardous contaminants, increasing local tax base, protecting greenfield sites, and slowing down urban sprawl [4,5]. Therefore, brownfield redevelopment plays a critical role in urban planning and development. In the redevelopment process, however, a large number of disputes inevitably arise due to a clash of interests and objectives among stakeholders involved consisting of local governments, developers, residents, and other decision-makers (DMs) [6,7,8]. For instance, environmental pollution produced during the process of land remediation may trigger some controversial issues between the developer and surrounding residents. There are two types of methodologies which can be employed for modeling and analyzing brownfield conflicts: the graph model for conflict resolution (GMCR), and conflict analysis approaches based on classical game theory.

GMCR is a highly powerful and flexible decision-making methodology for investigating challenging conflicts arising from social, economic, environmental, or other areas [9,10,11]. In classical game theory, cardinal preference such as utility values need to be determined [12]. The GMCR methodology, however, only requires relative preference information such as one state being less, equally, or more preferred to another, and it can also handle cardinal preference information. Moreover, the preference of a DM can be either transitive or intransitive. Specifically, if state *s*_1_ is more preferred to state *s*_2_ which is more preferred to *s*_3_ and, meanwhile, state *s*_1_ is more preferred to state *s*_3_, then the preference is said to be transitive. If state *s*_1_ is less or equally preferred to *s*_3_, then it is called intransitive. GMCR can be utilized to model and analyze a conflict having both transitive and intransitive preferences, whereas only transitive preference is allowed in game theory. Furthermore, a rich range of solution concepts or stabilities such as Nash stability [13], general metarationality (GMR) [14], symmetric metarationality (SMR) [14], and sequential stability (SEQ) [15,16] are formally defined within the GMCR paradigm for reflecting different behaviors of DMs in decision-making processes.

The preference elicitation is of great importance for investigating brownfield conflicts. In this research, a novel option weighing approach is proposed within the GMCR methodology for eliciting the true preference information of DMs involved in brownfield conflicts. Compared to existing preference ranking techniques, the preference obtained by using the new method is more objective and reasonable since states are ranked according to their similarities and closeness to the most and least preferred states instead of subjective option statements or weights assigned to options. Moreover, the proposed preference ranking technique is utilized to identify the actual preference of the local government and public in the brownfield conflict which occurred in China on 20 April 2016, which could provide more strategic insights for solving the land pollution dispute.

The rest of the paper is organized as follows: to begin with, a literature review regarding existing preference elicitation methods is presented in Section 2 within the framework of GMCR. In Section 3, the traditional option prioritization method based on the GMCR is briefly introduced. Then, a novel option weighing approach is developed for identifying the true preference information of DMs, followed by its detailed procedure. In Section 4, the novel approach is applied to a real-world brownfield conflict which occurred in China to demonstrate its advantages and applicability. Finally, conclusions are presented in Section 5.

## 2. Literature Review

### 2.1. Preference Ranking Methods Based on GMCR

For effectively eliciting the preference of DMs involved in a conflict, three preference ranking techniques were developed within the GMCR paradigm: direct ranking, option weighting, and option prioritization [17,18]. The option prioritization is the most useful and convenient approach to obtain a DM’s preference in terms of option statements, which was extended to handle various preference structures such as strength of preference [19,20,21], unknown preference [22,23,24], fuzzy preference [25,26,27,28], and gray preference [29,30]. To acquire three-level preference or strength of preference, Hou et al. [21] proposed a new option prioritization method by taking into account the degrees of preference regarding option statements. Yu et al. [24] then developed an option prioritization technique for ascertaining unknown preference, in which the uncertainty of preference was expressed by unknown option statements. By using fuzzy true values of option statements, Bashar et al. [31] put forward a fuzzy option prioritization method in order to determine fuzzy preference. Subsequently, gray option prioritization was developed by Zhao and Xu [32] for obtaining gray preference which was expressed by gray interval true values of option statements. To reflect the degrees of a DM’s confidence in option statements, Yin et al. [33] constructed a confidence-level function based on option prioritization.

The aforementioned research, however, was based on traditional option prioritization proposed by Fang et al. [17,18], in which the option statements are determined in terms of a DM’s subjective judgments or value-based system. The option statements provided by different individuals may be not the same, which indicates that the preference information using traditional option prioritization could differ from person to person.

### 2.2. The Use of GMCR in Brownfield Conflicts

As a highly flexible and powerful methodology, GMCR was utilized for analyzing many real-world disputes in brownfield redevelopment. Walker et al. [34] examined three conflicts in a private brownfield renovation in Kitchener, Canada, by considering decision-makers’ attitudes within the framework of GMCR. For addressing conflicts in brownfield reconstruction, Yousefi et al. [6] proposed a negotiation approach based on GMCR and then applied it to a real-life case of brownfield redevelopment located in southern Ontario, Canada. Kuang et al. [29] investigated a brownfield redevelopment conflict with the city government, property owner, and developer involved by using the gray-based GMCR. Philpot et al. [35] modeled a brownfield management dispute which occurred in Elmira, Ontario, Canada and analyzed the evolution of this conflict within a GMCR diagram. Yin et al. [33] applied an improved GMCR approach for modeling and analyzing a brownfield conflict regarding the Changzhou Foreign Languages School (CFLS) in China. Afterward, Yu and Pei [36] extended the brownfield conflict by considering the strength of preferences of DMs. However, the role of the public and the strategic delay of the local government were not taken into account in References [33,36].

Therefore, a new preference elicitation approach is purposefully proposed in this paper within the GMCR paradigm for effectively obtaining the true preference of DMs in brownfield development disputes. Moreover, the brownfield conflict related to the CFLS is systematically modeled and analyzed by using the novel developed preference ranking technique, in which the public and the local government’s strategic delay are considered.

## 3. Methodology

### 3.1. Graph Model and Traditional Option Prioritization Approach

A conflict can be expressed by a graph model $G=N,\;S,\;A,\;\left\{{\underset{˜}{≻}}_{i}:\;i\in N\right\}\rangle$, where *N*, *S*, and *A* represent the set of DMs, states, and oriented arcs, respectively, and $\left\{{\underset{˜}{≻}}_{i}:\;i\in N\right\}$ refers to a binary preference relationship. For instance, $s_{1}{≻}_{i}s_{2}$ means that state $s_{1}$ is more preferred to state $s_{2}$ by DM *i*, while $s_{1}{∼}_{i}s_{2}$ indicates that the two states are equally preferred by DM *i*. There are three preference elicitation approaches within the framework of the GMCR: direct ranking, option weighting, and option prioritization [17,18]. More specifically, the first approach can be easily used in smaller conflict models with limited states for directly ranking states according to a DM’s value-based system. In the second method, the ordering of states is determined according to their total weights which can be calculated by using each option’s weight. However, some preference ordering over states cannot obtained by using this approach. In a given conflict, if the number of options is *a*, then the number of states can be written as 2^*a*^ since each option can be chosen or not. The option prioritization is from the option level instead of the state level, which can significantly improve the computational efficiency of preference ranking, especially for large-scale conflicts. When analyzing the preference of DMs in a dispute, we only need to acquire a set of option statements for each DM which are ranked from most to least important in a hierarchical order. Therefore, the option prioritization approach is the most effective and convenient preference ranking method. An option statement consists of numbered options and logical operators.

In a given conflict, DM *i*’s preference over states can be intuitively expressed by a set of option statements $K_{i}=\left\{\Omega_{1}^{i},\;\Omega_{2}^{i},\ldots ,\;\Omega_{l}^{i},\ldots ,\;\Omega_{k}^{i}\right\}$ ordered by priority, where *k* is the total number of statements, and the former statement is more important than the latter one. Each option statement $\Omega_{l}^{i}\in K_{i}$ takes a truth value at state s∈S, denoted by $\Omega_{l}^{i}(s)$. Specifically, $\Omega_{l}^{i}(s)=T$ if the statement $\Omega_{l}^{i}(s)$ is true (T) at state *s*; otherwise, $\Omega_{l}^{i}(s)=F$. Let $\Psi_{l}^{i}(s)$ be the incremental score of $\Omega_{l}^{i}(s)$ at state *s*, where
(1)$$\Psi_{l}^{i}(s)=\left\{ \begin{array}{ll} 2^{k-l} & \mathrm{if}\;\Omega_{l}^{i}(s)=T\\ 0 & \mathrm{otherwise} \end{array}\right.$$


Then, the total score of state *s* can be determined by
(2)$$\Psi_{i}(s)=\sum_{l=1}^{k}\Psi_{l}^{i}(s)$$


All of the states can be ranked from most to least preferred according to their total scores, in which a state with a higher score is more preferred to a state with a lower score by DM *i*, and states with the same score are equally preferred. Therefore, each DM’s preference can be determined using the option prioritization approach. In many real-world conflicts, however, the most and least preferred states which can be easily and directly obtained may be not consistent with the results by using the option prioritization technique. Moreover, the option statements given by conflict analysts may be different or inconsistent because of subjective judgments. Hence, a novel preference ranking approach is purposefully developed below.

### 3.2. A Novel Option Weighing Technique for Preference Ranking

In a given conflict, as mentioned above, the most and least preferred states can be easily and intuitively identified by a focal DM, which indicates that it is not necessary to obtain them by using option prioritization. Moreover, the remaining states can be ranked from most to least preferred according to their similarities and closeness to the best and worst states, which is similar to the technique for order preference by similarity to an ideal solution (TOPSIS) [37,38]. In this research, a novel option weighing approach is proposed within the framework of GMCR for more efficiently eliciting the preference information of a DM. To begin with, some symbols which are used later, as well as their corresponding explanations, are presented in Table 1.

Let $O=\left\{o_{}^{1},\;o_{}^{2},\ldots ,\;o_{}^{l},\ldots ,\;o_{}^{k}\right\}$ be the set of options in a given conflict. A strategy for a DM is a combination of its options, and a state is an outcome when each DM’s strategy is determined, which can be represented by a *k*-dimensional column vector $s={\left(g_{}^{1},\;g_{}^{2},\ldots ,\;g_{}^{l},\ldots ,\;g_{}^{k}\right)}^{T}$, where
(3)$$g_{}^{l}=\left\{ \begin{array}{ll} Y & \mathrm{if}\;o_{}^{l}\;\mathrm{is}\;\mathrm{chosen}\\ N & \mathrm{otherwise} \end{array}\right.$$


Let the set of states be $S=\left\{s_{1},s_{2},\ldots ,s_{m}\right\}$. The procedure for determining a DM’s preference by using the novel option weighing approach is designed as follows:

**Step 1:** Identify the most and least preferred states for the DM denoted by $s_{{}^{+}}={\left(g_{+}^{1},\;g_{+}^{2},\ldots ,\;g_{+}^{l},\ldots ,\;g_{+}^{k}\right)}^{T}$ and $s_{{}^{-}}={\left(g_{-}^{1},\;g_{-}^{2},\ldots ,\;g_{-}^{l},\ldots ,\;g_{-}^{k}\right)}^{T}$, respectively.

**Step 2:** Determine the weight vector $W(s^{+})={\left(w_{1},\;w_{2},\ldots ,\;w_{l},\ldots ,\;w_{k}\right)}^{T}$ of all elements in $s_{{}^{+}}$, in which $w_{l}$ is the weight of $g_{+}^{l}$.
(1)Rank $w_{1},\;w_{2},\;\ldots ,\;w_{l},\;\ldots ,\;w_{k}$ in order of priority from most to least important, where
if $g_{+}^{l_{1}}\neq g_{-}^{l_{1}}$ and $g_{+}^{l_{2}}=g_{-}^{l_{2}}$, then $w_{l_{1}}>w_{l_{2}}$, andif $g_{+}^{l_{1}}{=g}_{+}^{l_{2}}$, then $w_{l_{1}}$ and $w_{l_{2}}$ are evaluated according to their importance to the focal DM.
(2)Assign the numerical scores for each weight according to their orders in (1), in which the *l*th biggest weight is 2^*k*^
^− *l*^.


Note that the weight vector represents the importance of each element in the most preferred state $s_{{}^{+}}$. Comparing the most and least preferred states, one can find that there are two types of elements in $s_{{}^{+}}$: one is the same as the corresponding element in $s_{{}^{-}}$; and the other is different from the corresponding element in $s_{{}^{-}}$. Note that the elements in the second type should be more important than those in the first type. Moreover, the elements in each type are ranked in terms of their importance to the focal DM.

**Step 3:** Calculate the true value matrix by comparing each state $s_{i}\in S$ with the best state $s_{{}^{+}}$ written by
(4)$$T=\left[\begin{array}{cccc} t_{11} & t_{12} & \cdot \cdot \cdot & t_{1k}\\ t_{21} & t_{22} & \cdot \cdot \cdot & t_{2k}\\ \cdot \cdot \cdot & \cdot \cdot \cdot & \cdot \cdot \cdot & \cdot \cdot \cdot \\ t_{m1} & t_{m2} & \cdot \cdot \cdot & t_{mk} \end{array}\right],$$
where the row represents $s_{{}^{+}}$, the column refers to states from $s_{1}$ to $s_{m}$, and
(5)$$t_{ij}=\left\{ \begin{array}{ll} 1 & \mathrm{if}\;g_{i}^{j}=g_{+}^{j}\\ 0 & \mathrm{otherwise} \end{array}\right.$$


**Step 4:** Normalize the true value matrix *T* using the below equation which is similar to that in TOPSIS method. The normalized matrix can be written by $A(a_{ij})$, in which
(6)$$a_{ij}=t_{ij}/\sqrt{\sum_{i=1}^{m}t_{ij}^{2}}$$


Then, the optimal and worst solution solutions can be determined as described below.

The positive ideal solution (PIS) is
(7)$$A^{+}=\left\{\max(a_{1j}^{},\;a_{2j}^{},\ldots ,\;a_{mj}^{})|j=1,\;2,\ldots ,\;k\right\}=\left\{a_{1}^{+},\;a_{2}^{+},\ldots ,\;a_{k}^{+}\right\}$$


Note that some normalized true values in the row of $s_{{}^{-}}$ could be the same as those in the row of $s_{{}^{+}}$. Unlike the traditional TOPSIS method, the negative ideal solution (NIS) in the novel option weighing approach is denoted by
(8)$$A^{-}=\left\{a_{1j}^{},a_{2j}^{},\cdot \cdot \cdot ,a_{kj}^{}\right\}=\left\{a_{1}^{-},a_{2}^{-},\cdot \cdot \cdot ,a_{k}^{-}\right\},$$
where *j* is the row of the least preferred state $s_{{}^{-}}$.

**Step 5:** Calculate the distance between every state $s_{i}\in S$ and the optimal and worst states as presented below.

The distance between $s_{i}$ and $A^{+}$ is
(9)$$D_{i}^{+}=\sqrt{\sum_{j=1}^{k}{\left[w_{j}(a_{j}^{+}-a_{ij}^{})\right]}^{2}},\;\mathrm{for}\;i=1,\;2,\;\ldots ,\;m.$$


The distance between $s_{i}$ and $A^{-}$ is
(10)$$D_{i}^{-}=\sqrt{\sum_{j=1}^{k}{\left[w_{j}(a_{j}^{-}-a_{ij}^{})\right]}^{2}},\;\mathrm{for}\;i=\;1,\;2,\;\ldots ,\;m.$$


**Step 6:** Calculate the closeness between every state $s_{i}\in S$ and the positive ideal solution $A^{+}$ by using the following equation:
(11)$$C_{i}=\frac{D_{i}^{-}}{D_{i}^{+}+D_{i}^{-}},\;\mathrm{for}\;i=1,\;2,\;\ldots ,\;m$$


**Step 7:** Determine the DM’s preference over states according to the values of $C_{i}$, in which a state with a higher value is more preferred to another state with a lower value. For example, if $C_{1}>C_{2}$, then state $s_{1}$ is more preferred to state $s_{2}$ ($s_{1}≻s_{2}$), and, if $C_{1}=C_{2}$, then states $s_{1}$ and $s_{2}$ are equally preferred ($s_{1}∼s_{2}$).

The detailed procedure for calculating a DM’s preference over states by using the novel option weighing method is depicted in Figure 1. As illustrated in Figure 1, the most and least preferred states should be firstly identified. By comparing the similarities and differences between the best and worst states, one can then construct the weight vector and true value matrix. Subsequently, the similarities and closeness of each state to the best and worst states should be analyzed, which is similar to the TOPSIS method. Finally, states can be ranked from most to least preferred according to their values of closeness to the optimal and worst solutions.

## 4. Application of GMCR for Conflict Resolution through a Case Study

### 4.1. Conflict Modeling

According to the China Central Television (CCTV) report on 20 April 2016, nearly 500 students of the Changzhou Foreign Languages School (CFLS), located in Changzhou, Jiangsu province, China, were diagnosed with serious health problems such as dermatitis, eczema, bronchitis, leukemia, and lymphoma after this school was relocated to its new campus in September 2015 [39,40,41]. The reason why children at CFLS fell sick was suspected to be air, soil, and water contamination by a land remediation program nearby, where the level of pollutants detected were excessively above safety standards. As shown in Figure 2, the new campus of CFLS in the blue area is very close to a brownfield remediation site in the red area, which was severely contaminated by three former chemical plants, Changlong, Changyu, and Huada. The CFLS pollution scandal attracted widespread attention and made headline news across China. Tens of thousands of people, including the parents of ill children and some environmentalists, flooded city streets to protest the environmental accident, prompting a joint intervention by the Ministry of Ecology and Environment of China (MEEC) and the Jiangsu Provincial Government.

On 26 August 2016, an environmental investigation report was released, which showed that the land remediation project failed to be completed on time before CFLS was relocated to its new campus. Moreover, it was the illegal operation during the remediation of the contaminated site that exposed toxic pollutants to air and caused the diseases of students. To reduce the threat of contaminations produced by the land remediation program to their children, many parents appealed to the local government to temporarily transfer the students at CFLS to a safe site. However, the local government insisted that the environmental quality around CFLS was normal and did not want to take action to transfer students. As a result, no agreement was reached by the two sides.

In this brownfield pollution conflict, there were mainly two decision-makers (DMs): the local government (LG) and the public (Public). LG had three options: (1) accelerate the land remediation project including environmental quality monitoring around CFLS; (2) transfer students at CFLS to a safe place temporarily to prevent them from being affected by air contamination, which would indicate that the environment around CFLS was indeed severely polluted; (3) choose to delay. The public, consisting of students’ parents and environmentalists, had one option: (4) whether or not to appeal for a public interest lawsuit.

A state is formed when each DM chooses its options. Since each option can be chosen (Y) or not (N), there are mathematically 2^4^ = 16 states. However, some states are infeasible and should be removed. For instance, it is impossible that LG chooses to both accelerate the land remediation project and delay. After infeasible states were excluded, there were ten feasible states left as given in Table 2, in which states are expressed by a column composed of “N” and “Y”. For example, state *s*_2_ (N N N Y) means that LG chose no options (N N N) while the public wanted to appeal for a public interest lawsuit (Y).

According to Table 2, the integrated graph model for describing state transitions in the brownfield conflict was drawn as shown in Figure 3, in which the nodes represent states and the solid and dashed lines indicate the state transitions of LG and the public, respectively. The labels on the arcs in Figure 3 refer to the DM who controls the movement among states. Moreover, an arc with double arrows indicates that the transition between states is reversible, whereas an arc with a single arrow means that the move is irreversible. As illustrated in Figure 3, for example, state *s*_8_ can unilaterally move to state *s*_10_, but not vice versa because LG cannot withdraw its decision once it chooses to transfer students at CFLS to a new site.

### 4.2. Preference Ranking

To demonstrate the advantage of the novel option weighing approach developed in this research, the traditional option prioritization method was firstly employed for eliciting the preference information of LG and the public in the brownfield conflict.

#### 4.2.1. Preference Ranking Using Traditional Option Prioritization

As explained in Section 2, the preference information of a DM can be expressed by a set of option statements according to the DM’s subjective judgments. In the brownfield dispute, there may exist many kinds of possible option statements for LG and Public since each conflict analyst has a unique understanding and subjective judgment regarding the dispute. Let us consider the option statements for LG and the public shown in Table 3.

By using the traditional option prioritization method [17,18] and the statements in Table 3, the preference information for LG and Public can be determined from the most preferred on the left to the least preferred on the right.
(12)$$\mathrm{LG}:\;s_{1}≻s_{7}≻s_{5}≻s_{9}≻s_{3}≻s_{2}∼s_{4}≻s_{8}≻s_{6}≻s_{10}.$$
(13)$$\mathrm{Public}:\;s_{9}≻s_{10}≻s_{5}≻s_{6}≻s_{8}≻s_{7}≻s_{2}≻s_{1}≻s_{4}≻s_{3}.$$


From Equation (12), one can find that state *s*_3_ is less preferred to states *s*_7_, *s*_5_, and *s*_9_ by LG in the brownfield conflict, which is contradictory to the fact that LG does not wants to accelerate the land remediation project and transfer the students. Furthermore, state *s*_2_ is equally preferred to state *s*_4_ in Equation (12). In the brownfield conflict, however, strategic delay is not a good choice for LG when the public decides to file a public interest litigation because the environmental investigation report released by the Ministry of Ecology and Environment of China and the Jiangsu provincial government on 26 August 2016 showed that it was the environmental pollution during the land remediation that caused the illness of the students at CFLS. Therefore, the preference information of LG obtained by using traditional option prioritization is inconsistent with the real situation. The main reason is that the option statements are given according to personal judgment, and different conflict analysts may present various sets of option statements.

#### 4.2.2. Preference Ranking Using the Novel Option Weighing Approach

To effectively elicit the true preference information of LG and the public, the novel option weighing method developed in Section 3 was utilized following a detailed procedure.

**Step 1:** Identify the most and least preferred states of LG and the public in the brownfield pollution conflict.

For LG, its most preferred state is *s*_1_ (N N N N), in which LG takes no action and the public does not appeal. Its least preferred state is *s*_10_ (Y Y N Y), where LG speeds up the land remediation and takes action to transfer students to a safe place while the public appeals for a public interest lawsuit. For the public, its most and least preferred states are *s*_9_ (Y Y N N) and *s*_3_ (N N Y N), respectively. Specifically, state *s*_9_ means that LG accelerates the remediation of contaminated land and temporarily transfers children, and the public does not appeal, which is the optimal solution for the public. State *s*_3_ is the worst situation for the public since it does not appeal when LG adopts a strategic delay.

**Steps 2:** Determine the weight vector and true value matrix for LG and the public.

To begin with, the optimal and worst states for LG are denoted by *s*_1_ (N^1^ N^2^ N^3^ N^4^) and *s*_10_ (Y^1^ Y^2^ N^3^ Y^4^), in which the superscript refers to the order. By comparing *s*_1_ and *s*_10_, one can find that the elements of N^1^, N^2^, and N^4^ in *s*_1_ are different from those in *s*_10_, whereas N^3^ is the same. As illustrated in Section 3, the different elements are more important than the same elements. Hence, the weights for N^1^, N^2^, and N^4^ are bigger than the weight for N^3^, denoted by *w* (N^1^, N^2^, N^4^) > *w* (N^3^). Furthermore, one can deduce that *w* (N^4^) > *w* (N^2^) > *w* (N^1^). In conclusion, the following relationship can be determined: *w* (N^4^) > *w* (N^2^) > *w* (N^1^) > *w* (N^3^).

According to Step 2 in Section 3, *w* (N^4^) = 2^3^, *w* (N^2^) = 2^2^, *w* (N^1^) = 2^1^, and *w* (N^3^) = 2^0^. By comparing each state with the most preferred state *s*_1_, one can obtain the true values of states for LG as shown in Table 4.

As shown in Table 4, the first row stands for the most preferred states *s*_1_ (N^1^ N^2^ N^3^ N^4^) by LG with the corresponding weight in brackets. Moreover, the numbers 0 and 1 in Table 4 refer to true values, in which “1” means that an element of a particular state in the row is the same as that in the best state *s*_1_. For example, the true values in the fifth row for state *s*_4_ (**N N** Y Y) are “1 1 0 0” because only the former two elements of *s*_4_ in bold are the same as those in *s*_1_ (**N^1^ N**^2^ N^3^ N^4^).

Similarly, the weight vector and true value matrix for the public can be acquired as given in Table 5, in which the most preferred state for the public is *s*_9_ (Y^1^ Y^1^ N^1^ N^1^) and the weights of elements in *s*_9_ are 2^1^, 2^3^, 2^2^, and 2^0^, respectively.

**Step 3:** Calculate the values of *C_i_* of each state and rank states from most to least preferred.

By using Equations (4)–(11), the values of *C*_LG_ and *C*_Public_ can be determined as summarized in Table 6.

According to the values of *C*_LG_ and *C*_Public_ in Table 6, the preference information of LG and the public can be easily determined.
(14)$$\mathrm{LG}:\;s_{1}≻s_{3}≻s_{7}≻s_{5}≻s_{9}≻s_{2}≻s_{4}≻s_{8}≻s_{6}≻s_{10}.$$
(15)$$\mathrm{Public}:\;s_{9}≻s_{10}≻s_{5}≻s_{6}≻s_{8}≻s_{7}≻s_{2}≻s_{1}≻s_{4}≻s_{3}.$$


Comparing Equations (14) and (15) and Equations (12) and (13), one can draw the following two conclusions:

(1) The preference of LG by using the novel option weighing approach is more reasonable and accurate than that obtained by traditional option prioritization since *s*_3_ is more preferred to states *s*_7_, *s*_5_, and *s*_9_, and *s*_2_ is more preferred to *s*_4_ by LG in Equation (14).

(2) The preference of the public by using the proposed option weighing approach is the same as that obtained by traditional option prioritization, which indicates that the new preference ranking approach proposed in this research can obtain identical results to traditional option prioritization.

In the traditional option prioritization technique, the option statements are given according to a DM’s subjective judgments, and different DMs may provide various sets of option statements, which indicates that the preference ranking of states by using option prioritization could differ from person to person. In the novel option weighing approach, states are ranked in terms of their similarities and closeness to the most and least preferred states. The preference of a DM is uniquely determined once the optimal and worst states are identified. Therefore, the new preference ranking technique proposed in this research is more objective and credible in comparison with existing option prioritization methods.

### 4.3. Stability Analyses

For portraying strategic moves and countermoves among DMs in a conflict, four classical stabilities or solution concepts are formally defined within the GMCR paradigm, namely, Nash, GMR, SMR, and SEQ [9,10,11]. Let the set of DMs and states in a conflict be *N* and *S*, respectively. The definitions for the aforementioned four stabilities are briefly introduced as shown in Table 7.

As illustrated in Table 7, each stability indicates a specific decision-making behavior of the focal DM, and DMs in a conflict could adopt different stabilities [42]. GMCR II is a flexible and powerful decision support system for conveniently implementing conflict modeling and analysis [17,18]. With DMs, options, and preference input into GMCR II, the results of stability analyses for the brownfield conflict can be determined as presented in Table 8, in which a check “√” under a DM at a given state indicates that the state is stable for the DM under the corresponding solution concept, and “L”, “P”, and “E” stand for LG, Public, and equilibrium, respectively. If a state is stable for each DM in a conflict, it is called an equilibrium or resolution denoted by an asterisk “*”, which can provide meaningful strategic insights for addressing the conflict.

According to Table 8, the strong equilibria, stable for all DMs under all of the Nash, GMR, SMR, and SEQ stabilities, are states *s*_2_ and *s*_5_. State *s*_2_ means that LG takes no actions and the public appeals for a public interest lawsuit, which is consistent with the fact that two environmental non-governmental organizations, consisting of Friends of Nature and the China Biodiversity Conservation and Green Development Foundation, filed public interest litigation on 26 April 2016 [41]. State *s*_5_ indicates that LG agrees to transfer the students of the Changzhou Foreign Languages School to a safe site, and the public does not appeal, which is a satisfactory equilibrium stable for both parties. This solution indicates that LG should take into account the public opinion when making strategic actions. At the initial period of this dispute, the public wants LG to temporarily relocate CFLS to avoid the air pollution caused by the land remediation project. If LG satisfied this requirement, the dispute between LG and the public could be alleviated.

## 5. Conclusions

A novel option weighing approach was developed in this paper for eliciting the actual preference information of a DM involved in a brownfield conflict. In the proposed preference ranking approach, states are ranked according to their similarities and closeness to the most and least preferred states, which is more objective and reasonable than traditional option prioritization. Subsequently, a real-world brownfield conflict regarding the Changzhou Foreign Languages School in China was systematically modeled and analyzed to show how the new preference ranking method can be employed for determining DMs’ true preference. The case study demonstrates that (1) the local government’s preference obtained by the novel option weighing approach was more reasonable and accurate than that by using traditional option prioritization; (2) the preferences of the public by employing the proposed preference ranking method and traditional option prioritization were the same; (3) state *s*_5_ was a satisfactory equilibrium, in which meaningful and strategic insights were discovered for addressing the brownfield conflict.

In the future, the novel option weighing approach can be extended for handling the strength of preference and unknown or fuzzy preference. Furthermore, the brownfield conflict model developed in this paper could be extended by considering more DMs or the coalitions among them. Considering that many multiple-criteria decision-making techniques were employed in brownfield redevelopment projects [3,39,43], the weight vector in the proposed preference ranking method could be determined using multi-criteria decision analysis (MCDA) methods.

## Figures and Tables

**Figure 1 ijerph-16-04088-f001:**
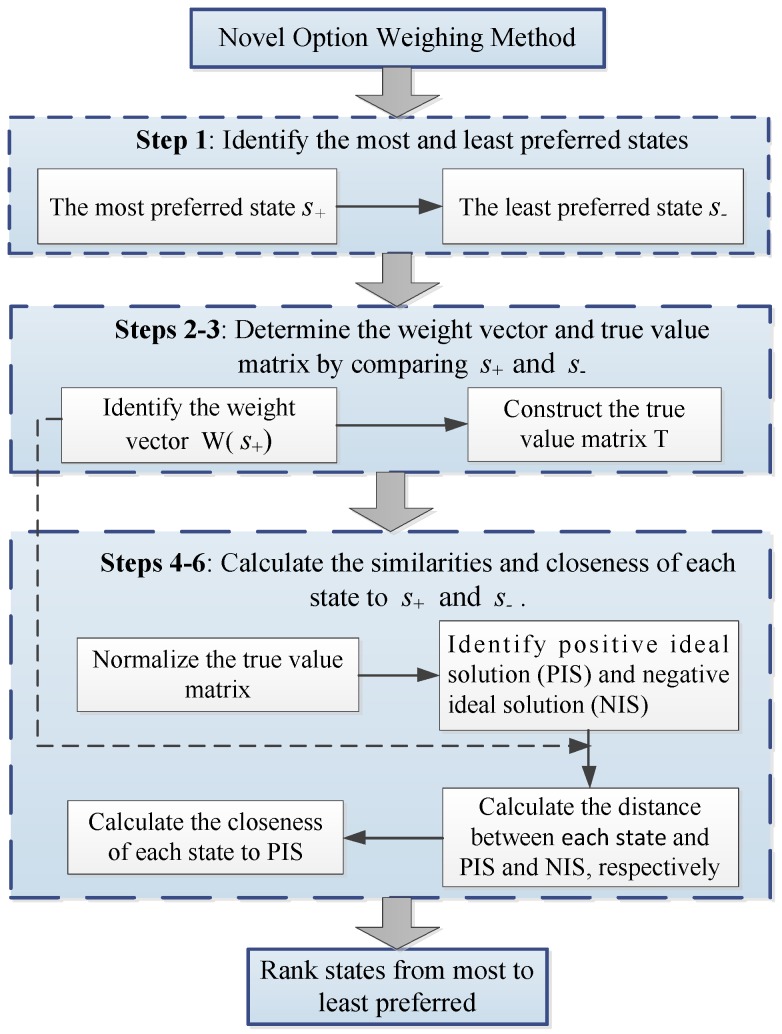
The procedure for preference ranking using the novel option weighing approach.

**Figure 2 ijerph-16-04088-f002:**
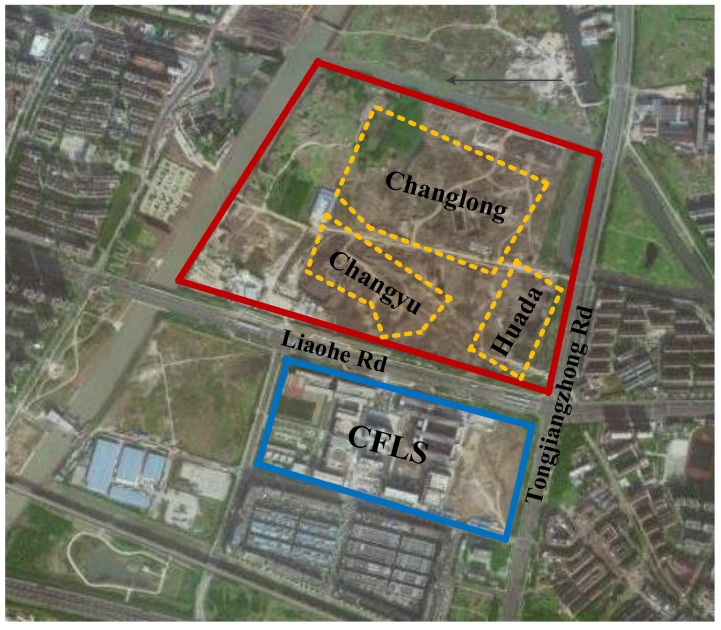
The new campus of Changzhou Foreign Languages School (CFLS) next to a contaminated brownfield site.

**Figure 3 ijerph-16-04088-f003:**
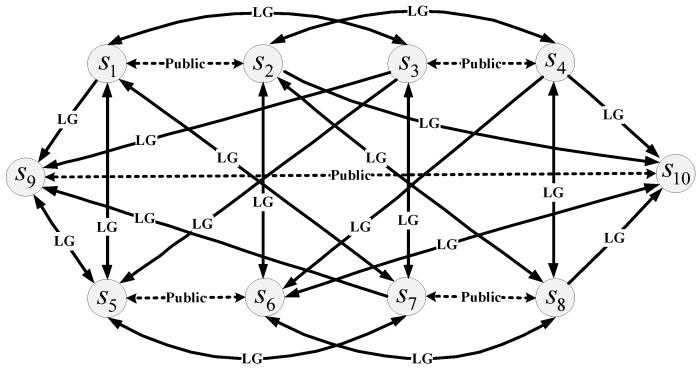
Integrated graph model for the brownfield dispute.

**Table 1 ijerph-16-04088-t001:** Symbols and their explanations.

Symbol	Explanation	Symbol	Explanation
*S*	The set of states	*O*	The set of options
*s*	A particular state	*o*	A particular option
*m*	The number of states	*k*	The number of options
*g^l^*	Whether or not the *l*th option is chosen	*S* _+_	The most preferred state
*S* _−_	The least preferred state	*W*	Weight vector
$w_{l}$	The weight of $g_{+}^{l}$	*T*	True value matrix
*t_ij_*	The entry (*i*, *j*) in *T*	*A*(*a_ij_*)	Normalized true value matrix
*a_ij_*	The entry (*i*, *j*) in *A*	**A** ^+^	The positive idea solution
**A** ^−^	The negative ideal solution		

**Table 2 ijerph-16-04088-t002:** Decision-makers (DMs), options, and feasible states in the brownfield conflict. LG—local government; Y—yes; N—no.

DMs	Options	*s* _1_	*s* _2_	*s* _3_	*s* _4_	*s* _5_	*s* _6_	*s* _7_	*s* _8_	*s* _9_	*s* _10_
LG	1. Accelerate	N	N	N	N	N	N	Y	Y	Y	Y
	2. Transfer	N	N	N	N	Y	Y	N	N	Y	Y
	3. Delay	N	N	Y	Y	N	N	N	N	N	N
Public	4. Appeal	N	Y	N	Y	N	Y	N	Y	N	Y

**Table 3 ijerph-16-04088-t003:** Option statements of LG and Public.

DMs	Statements	Explanation
LG	-4	LG wants the public not to appeal
	-3IF-4	LG will not delay the issue if the public does not appeal
	-2	LG does not want to transfer the students to a safe site since it denied the fact that the environment around CFLS was severely polluted
	-1	LG does not want to accelerate the land remediation project due to increasing costs
Public	2	Public wants LG to transfer the students to a safe site
	-3	Public does not want LG to delay
	1	Public wants LG to accelerate the land remediation project
	-4IFF2	Public will not appeal if and only if LG chooses to transfer the students to a safe site

**Table 4 ijerph-16-04088-t004:** True values of states for LG.

	Weight	N^1^ (2^1^)	N^2^ (2^2^)	N^3^ (2^0^)	N^4^ (2^3^)
States					
*s* _1_		1	1	1	1
*s* _2_		1	1	1	0
*s* _3_		1	1	0	1
*s* _4_		1	1	0	0
*s* _5_		1	0	1	1
*s* _6_		1	0	1	0
*s* _7_		0	1	1	1
*s* _8_		0	1	1	0
*s* _9_		0	0	1	1
*s* _10_		0	0	1	0

**Table 5 ijerph-16-04088-t005:** True values of states for the public.

	Weight	Y^1^ (2^1^)	Y^2^ (2^3^)	N^3^ (2^2^)	N^4^ (2^0^)
States					
*s* _1_		0	0	1	1
*s* _2_		0	0	1	0
*s* _3_		0	0	0	1
*s* _4_		0	0	0	0
*s* _5_		0	1	1	1
*s* _6_		0	1	1	0
*s* _7_		1	0	1	1
*s* _8_		1	0	1	0
*s* _9_		1	1	1	1
*s* _10_		1	1	1	0

**Table 6 ijerph-16-04088-t006:** Values of *C*_LG_ and *C*_Public_.

States	*C* _LG_	*C* _Public_
*s* _1_	1.000	0.255
*s* _2_	0.338	0.263
*s* _3_	0.919	0.000
*s* _4_	0.341	0.093
*s* _5_	0.692	0.809
*s* _6_	0.172	0.796
*s* _7_	0.828	0.302
*s* _8_	0.308	0.307
*s* _9_	0.662	1.000
*s* _10_	0.000	0.907

**Table 7 ijerph-16-04088-t007:** Four kinds of stabilities in the graph model for conflict resolution (GMCR). GMR—general metarationality; SMR—symmetric metarationality; SEQ—sequential stability.

Stability	Definition
Nash	State s∈S is Nash stable for DM i∈N if the DM cannot move to more preferred states from *s*.
GMR	State s∈S is GMR stable for DM i∈N if its opponents can prevent the DM from moving to any more preferred states regardless of their preference when sanctioning.
SMR	State s∈S is SMR stable for DM i∈N if the DM has no chance to escape from the sanctions by its opponents regardless of their preference.
SEQ	State s∈S is SEQ stable for DM i∈N if its opponents can prevent the DM from moving to any more preferred states by levying only unilateral improvements when sanctioning.

**Table 8 ijerph-16-04088-t008:** The outcomes of stability analyses for the brownfield conflict. L—local government; P—public; E—equilibrium.

States	Nash	GMR	SMR	SEQ
L	P	E	L	P	E	L	P	E	L	P	E
*s* _1_	√			√	√	*	√	√	*			
*s* _2_	√	√	*	√	√	*	√	√	*	√	√	*
*s* _3_				√			√			√		
*s* _4_		√			√			√			√	
*s* _5_	√	√	*	√	√	*	√	√	*	√	√	*
*s* _6_	√			√			√			√		
*s* _7_				√	√	*	√	√	*	√	√	*
*s* _8_		√			√			√			√	
*s* _9_		√		√	√	*	√	√	*		√	
*s* _10_					√			√			√

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
