# Peer review of "A Novel Preference Elicitation Technique Based on a Graph Model and Its Application to a Brownfield Redevelopment Conflict in China"

_ijerph, 2019, doi:10.3390/ijerph16214088_

Round 1

Reviewer 1 Report

General Comment:

A novel preference ranking technique is presented in this research for effectively determining a decision maker’s preference in a given conflict within the framework of the graph model for conflict resolution (GMCR). The preference information by using the improved preference elicitation approach is more straightforward and objective than traditional option prioritization methods. The whole application background of this new method in this paper is limited in brownfields disputes, whereas the proposed approach can be widely used in many more areas. The author can highlight it to reflect your contribution. To sum up, the manuscript is well presented and organized and, hence, I would like to recommend that the paper be accepted for publication in IJERPH with a minor revision by considering the following comments.

Detailed Comments:

Introduction

The author should provide the rationale about how the conflict occurs between different stakeholders involved in the brownfield redevelopment projects (BRPs)? Especially, at the end of first paragraph, and then author should introduce the methodology to solve the conflict in BRPs.

Literature review     

It is not clear in the literature review section that specifically what kind of literature the author wants to describe? Either is it related to methodology or brownfield redevelopment? The author should give the sub-heading under the literature review section to make it clearer. The author missed the literature review on brownfield redevelopment and its application in developing countries, especially in China. Further, author missed the literature on conflict occurrence in BRPs. Therefore, author can provide in-depth literature review in three perspectives: 1) Brow filed redevelopment & China; 2) Conflict occurrence in BRPs; 3) Use of GMCR for resolving conflicts in BRPs.

Application

Section 4, application: A brownfield conflict in China. The author should modify the heading. As, this study is based on a case study of CFLA, Changzhou, China. Therefore, author should restructure the heading as “Application of GMCR for conflict resolution through a case study”

In section 4, the author should describe more details about the State S5 as author indicated that this is satisfactory solution for both parties. The author should discuss the result in light of earlier studies.

Some grammar errors exist in the manuscript. Therefore, you need to carefully check it again. Several examples are shown for you: the “play” in line 12 should be “plays”; the “existing the traditional” should be “the traditional existing”; the “represent” in line 156 should be “represents”; the “is” in line 255 should be “was”; the “to eliciting” in line 339 should be “for eliciting”. Page 3, lines 104-106, the authors should briefly explain the concept of the binary preference relationship, direct ranking and option weighting. Page 3, the following two articles related on option prioritization approach should be cited in Section 3.1.

[1] Fang, L.; Hipel, K.W.; Kilgour, D.M.; Peng, X. A decision support system for interactive decision making—Part I: Model formulation. IEEE Trans. Syst. Man Cybern. 2003, 33, 42–55.
[2] Fang, L.; Hipel, K.W.; Kilgour, D.M.; Peng, X. A decision support system for interactive decision making— Part II: Analysis and output interpretation. IEEE Trans. Syst. Man Cybern. 2003, 33, 56–66.

Page 4, lines 149-153, the authors mentioned that the weights of different elements between s+ and s- are bigger than those of identical elements. The elements in s+ are divided into two types. They should also briefly explain how to evaluate the weights of elements of each type as given in line 146. Page 6, line 204, “LG” should be changed by “the local government” since it is used for the first time. Page 13, lines 368-369, the format of titles in [4], [39] and [40] should be the same as others.

Reviewer 2 Report

Dear Authors,

here is my response to the manuscript entitled “A Novel Preference Elicitation Technique Based on Graph Model and Its Application to a Brownfield Redevelopment Conflict in China”.

Comments on abstract, title, references

The paper proposes a novel preference ranking technique within the graph model for conflict resolution paradigm to effectively and accurately garner DMs’ actual preferences, in which states are ranked according to their similarities and closeness to the most and least preferred states instead of subjective option statements or weights in traditional preference ranking methods.

The aim of the work is clearly described in the abstract. The title is informative and relevant. The methodology is outlined.

Comments on introduction/background

The context of the research is clearly established and the literature review seem to be sufficient. The research gap is also outlined.

Line 81-83 – incorrect font size – to correct.

Comments on methodology

The study methods are valid, and the description contain enough detail in order to replicate the study.

Interesting novel option weighing approach.

Comments on results

The data is presented in an appropriate way. Tables and figures relevant and clearly presented.     Titles, columns, and rows labelled correctly and clearly.

Comments on discussion and conclusions

Very good and interesting paper therefore I recommend to accept it after slight editing revision.

Reviewer 3 Report

The paper presents a novel approach based on graph model for conflict resolution in brown field redevelopment area in China

The paper is clearly written and the well presented.

The references are appropriated to the methodology presented and the case study

For this reason the paper can be publish in the present form considering some minor revision

Introduction

The author presents the Graph Model for conflict resolution compared to others elicitation model. I suggest also to integrate the references with the litterature on multicriteria methods with particular reference to Saaty (AHP) and Munda (NAIADE Methods) often used in redevelopment project, in the Italian context. See for examples

Vincenzo Del Giudice & Pierfrancesco De Paola & Torrieri Francesca & Peter J. Nijkamp & Aviad Shapira, 2019. "Real Estate Investment Choices and Decision Support Systems," Sustainability, MDPI, Open Access Journal, vol. 11(11), pages 1-18, June.

Methodology

The author at line 106-107 state that “ option weighting and option prioritization is the most 
 effective and convenient preference ranking method “. 
 Please motivate more this statement

Line 157 Step 4 Please indicate why you use this normalization rules compared to the others

Case study

The author consider only two actors in the choice problem. How many actors can be included in the model. How can be evaluate possible coalition among different group of actors?
